# REDUNDANCY-FREE COMPUTATION GRAPHS FOR GRAPH NEURAL NETWORKS

## ABSTRACT

Graph Neural Networks (GNNs) are based on repeated aggregations of information across nodes' neighbors in a graph. However, because common neighbors are shared between different nodes, this leads to repeated and inefficient computations. We propose *Hierarchically Aggregated computation Graphs* (HAGs), a new GNN graph representation that explicitly avoids redundancy by managing intermediate aggregation results hierarchically, and eliminating repeated computations and unnecessary data transfers in GNN training and inference. We introduce an accurate cost function to quantitatively evaluate the runtime performance of different HAGs and use a novel search algorithm to find optimized HAGs. Experiments show that the HAG representation significantly outperforms the standard GNN graph representation by increasing the end-to-end training throughput by up to $2.8\times$ and reducing the aggregations and data transfers in GNN training by up to $6.3\times$ and $5.6\times$. Meanwhile, HAGs improve runtime performance by preserving GNN computation, and maintain the original model accuracy for arbitrary GNNs.

## 1 INTRODUCTION

Graph neural networks (GNNs) have shown state-of-the-art performance across a number of tasks with graph-structured data, such as social networks, molecule networks, and webpage graphs (Kipf and Welling, 2016; Hamilton et al., 2017; Ying et al., 2018; Xu et al., 2019). GNNs use a recursive neighborhood aggregation scheme — in a GNN layer, each node aggregates its neighbors' activations from the previous GNN layer and uses the aggregated value to update its own activations. The activations of the final GNN layer are used for downstream prediction tasks, such as node classification, graph classification, or link prediction.

Due to the clustering nature of real-world graphs, different nodes in a graph may share a number of common neighbors. For example, in webpage graphs, different websites under the same domain generally have a number of common links (i.e., neighbors). As another example, in recommender systems, users in the same group may have interests in common items.

However, existing GNN representations do not capture these common neighbors in real-world graphs, leading to redundant and unnecessary computation in both GNN training and inference. In particular, most existing GNN models (Kipf and Welling, 2016; Xu et al., 2019; Wu et al., 2019) use the full-batch training method that computes the activations for all nodes in each layer. Existing GNN representations use a *computation graph* (referred to as GNN-graph) to define computation in a GNN layer. The GNN-graph includes a tree structure for each node $v$ in the input graph describing how to compute $v$'s activations by aggregating the previous-layer activations of $v$'s neighbors. Figure 1b shows the GNN-graph of the input graph in Figure 1a; for example, for node $A$, its neighbor's activations $h_B^{(k-1)}$, $h_C^{(k-1)}$ and $h_D^{(k-1)}$ from the layer $k-1$ are aggregated to compute new activations $h_A^{(k)}$ for the layer $k$ (see the top portion of Figure 1b). The new activations of the other nodes are computed similarly using the previous activations of their neighbors. Notice that this representation results in redundant computation and data transfers. In this small example, both $\{A, B\}$ and $\{C, D\}$ are aggregated twice. In wider and multi-layer GNNs, the redundancies in existing GNN representations account for a significant fraction of all computation. For example, our experiments show that in modern GNNs up to 84% of the aggregations are redundant and avoidable.

In this paper, we propose a new GNN representation called *Hierarchically Aggregated computation Graphs* (HAGs). Figure 1c shows one possible HAG for the input graph in Figure 1a. HAGs are

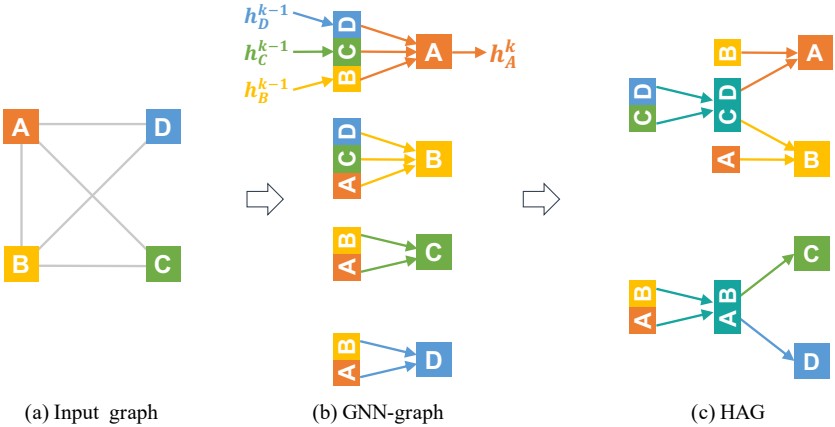

Figure 1: Comparison between a GNN-graph and an equivalent HAG. (a) Input graph; (b) 1-layer GNN computation graph (GNN-graph); (c) HAG that avoids redundant computation. The GNN-graph computes new activations $h_v^{(k)}$ by aggregating the previous-layer activations of $v$'s neighbors. Because nodes in the input graph share common neighbors, the GNN-graph performs redundant computation (e.g., both $\{A, B\}$ and $\{C, D\}$ are aggregated twice). (c) By identifying common computational patterns, the HAG avoids repeated computation.

functionally equivalent to standard GNN-graphs (produce the same output), but represent common neighbors across different nodes using aggregation hierarchies, which eliminates redundant computation and unnecessary data transfers in both GNN training and inference. In addition, a HAG is agnostic to any particular GNN model, and can be used to eliminate redundancy for arbitrary GNNs.

For a GNN-graph, there exist numerous equivalent HAGs with different aggregation hierarchies and runtime performance. We introduce an accurate cost function to quantitatively estimate the performance of different HAGs and develop a novel HAG search algorithm to automatically find optimized HAGs.

Theoretically, we prove that the search algorithm finds HAGs with strong performance guarantees: (1) for GNN models whose neighborhood aggregations require a specific ordering on a node's neighbors, the algorithm finds a *globally optimal* HAG under the cost function; and (2) for other GNN models, the algorithm finds HAGs whose runtime performance is at least a $(1 - 1/e)$ approximation ($\approx 63\%$) of globally optimal HAGs using the submodularity property (Mossel and Roch, 2007). Empirically, the algorithm finds HAGs that lead to major improvements, reducing the runtime by up to $2.9\times$.

Our HAG abstraction maintains the predictive performance of GNNs but leads to much faster training and inference. We evaluate the performance of HAGs on five real-world datasets and along three dimensions: (a) end-to-end training and inference runtime; (b) number of aggregations; and (c) size of data transfers. Experiments show that HAGs increase the end-to-end training and inference performance by up to $2.8\times$ and $2.9\times$, respectively. In addition, compared to GNN-graphs, HAGs reduce the number of aggregations and the size of data transfers by up to $6.3\times$ and $5.6\times$, respectively.

## 2 RELATED WORK

**Graph neural networks** have been used to solve various real-world tasks with relational structures (Kipf and Welling, 2016; Hamilton et al., 2017; Ying et al., 2018; Xu et al., 2019). Fast-GCN (Chen et al., 2018) and SGC (Wu et al., 2019) accelerate GNN training using importance sampling and removing nonlinearilities. This paper solves an orthogonal problem: how to optimize GNN efficiency while maintaining network accuracy. Our methods can be applied automatically to eliminate redundancy for arbitrary GNN models.

**Join-trees** are a tree decomposition technique that maps a graph into a corresponding tree structure to solve optimization problems on the graph, such as query optimization (Flum et al., 2002). Although a join-tree provides a possible way to find optimal HAGs for a GNN-graph, its time complexity is exponential in the *treewidth* of a graph (Arnborg et al., 1987), and real graphs tend to have very large treewidths. For example, Adcock et al. (2016) shows that the treewidth of real-world social networks grows linearly with the network size, making it infeasible to use join-trees to find optimal HAGs.

Table 1: Existing GNNs described in our abstraction. GraphSAGE-P and GraphSAGE-LSTM are the pooling and LSTM variants of GraphSAGE, respectively. $\sigma$ and *max* indicate element-wise non-linear activation and max functions. For sequential AGGREGATE, $v_i$ denotes the $i$-th in-neighbor of node $v$.

| GNN | AGGREGATE($\{h_u^{(k-1)}\|u \in \mathcal{N}(v)\}$) | UPDATE($a_v^{(k)}, h_v^{(k-1)}$) |
|---|---|---|
| | Set AGGREGATE | |
| GCN (Kipf and Welling, 2016) | $a_v^{(k)} = \sum_{u \in \mathcal{N}(v)} h_u^{(k-1)}$ | $h_v^{(k)} = \sigma(W^{(k)} \cdot \frac{a_v^{(k)} + h_v^{(k-1)}}{|\mathcal{N}(v)|+1})$ |
| GIN (Xu et al., 2019) | $a_v^{(k)} = \sum_{u \in \mathcal{N}(v)} h_u^{(k-1)}$ | $h_v^{(k)} = \sigma\left(W \cdot \left((1 + \epsilon^{(k)})h_v^{(k-1)} + a_v^{(k)}\right)\right)$ |
| | Sequential AGGREGATE | |
| GCN-LSTM (Hamilton et al., 2017) | $a_v^{(k)} = LSTM(h_{v_1}^{(k-1)}, ..., h_{v_\mathcal{N}}^{(k-1)})$ | $h_v^{(k)} = \sigma\left(W^{(k)} \cdot (a_v^{(k)}, h_v^{(k-1)})\right)$ |
| $N$-ary Tree-LSTM (Tai et al., 2015) | $a_v^{(k)} = Tree\text{-}LSTM\text{-}Agg(h_{v_1}^{(k-1)}, ..., h_{v_\mathcal{N}}^{(k-1)})$ | $h_v^{(k)} = Tree\text{-}LSTM\text{-}Update(a_v^{(k)}, h_v^{(k-1)})$ |

**Computation reduction in DNNs.** Several techniques have been proposed to reduce computation in DNNs, including pruning weights (Han et al., 2015) and quantization (Han et al., 2016). These techniques reduce computation at the cost of modifying networks, resulting in decreased accuracy (as reported in these papers). By contrast, we propose a new GNN representation that accelerates GNN training by eliminating redundancy in GNN-graphs while maintaining the original network accuracy.

## 3 HIERARCHICALLY AGGREGATED COMPUTATION GRAPHS (HAGS)

**Existing GNN-graph representation.** An input graph $\mathcal{G} = (\mathcal{V}, \mathcal{E})$ has nodes $\mathcal{V}$ and edges $\mathcal{E}$. For each node $v \in \mathcal{V}$, $\mathcal{N}(v)$ denotes the set of neighbors of $v$, and $x_v$ denotes the input node features. A GNN iteratively learns representations for individual nodes over the entire graph through a number of GNN layers, as shown in Algorithm 1. The learned activations of node $v$ at layer $k$ is $h_v^{(k)}$, and we initialize $h_v^{(0)}$ with $x_v$. At the $k$-th layer, $a_v^{(k)}$ denotes the aggregated activations of $v$'s

---

**Algorithm 1** An abstraction for GNNs. $\mathcal{V}$ is the set of nodes in an input graph, and $\mathcal{N}(v)$ denotes the set of neighbors for node $v$.

1: $h_v^{(0)} = x_v, \forall v \in \mathcal{V}$
2: **for** $k = 1$ to $K$ **do**
3:     **for** $v \in \mathcal{V}$ **do**
4:         $a_v^{(k)} \leftarrow$ AGGREGATE($\{h_u^{(k-1)}|u \in \mathcal{N}(v)\}$)
5:         $h_v^{(k)} \leftarrow$ UPDATE($a_v^{(k)}, h_v^{(k-1)}$)
6: **Goal:** minimize $\mathcal{L}(\{h_v^{(K)}|v \in \mathcal{V}\})$

---

neighbors, which is combined with $h_v^{(k-1)}$ to compute an updated activation $h_v^{(k)}$. The learned node activations of the final layer (i.e., $h_v^{(K)}$) are used for downstream learning tasks, and a GNN model generally minimizes a loss function $\mathcal{L}$ that takes the final node activations as inputs (line 6).

Existing GNN models use a GNN *computation graph* (GNN-graph) to describe the computation in each GNN layer, as shown in Figure 1b. For each node $v$ in the input graph, the GNN-graph includes an individual tree structure to define how to compute the activations $h_v^{(k)}$ of node $v$ by aggregating the previous-layer activations of $v$'s neighbors (i.e., $\{h_u^{(k-1)}, u \in \mathcal{N}(v)\}$). GNN-graphs are efficient at expressing direct neighborhood relations between nodes, but are not capable of capturing common neighbors across multiple nodes, leading to redundant computation in GNN training and inference.

### 3.1 HAG REPRESENTATION

We propose a new graph representation called *Hierarchically Aggregated computation Graphs* (HAGs) for GNNs. HAGs eliminate redundancy in the GNN-graph representation by hierarchically managing and reusing intermediate aggregation results. A HAG $\widehat{\mathcal{G}} = (\widehat{\mathcal{V}}, \widehat{\mathcal{E}})$ has nodes $\widehat{\mathcal{V}} = \mathcal{V} \cup \mathcal{V}_A$ and edges $\widehat{\mathcal{E}}$, where $\mathcal{V}$ is the set of nodes in the original graph, and $\mathcal{V}_A$ is a new set of *aggregation* nodes. Each aggregation node in $\mathcal{V}_A$ represents the intermediate aggregations result for a subset of nodes (i.e., aggregation on a subset of $h_v^{(k-1)}$). For the HAG example in Figure 1c, the new nodes *AB* and *CD* denote the aggregation results of $\{A, B\}$ and $\{C, D\}$, respectively. A HAG can contain a multi-level aggregation hierarchy. For example, Figure 1c can also have a third aggregation node *ABCD* that depends on *AB* and *CD*. Similar to edges in GNN-graphs, an edge $(u, v)$ in a HAG denotes an aggregation relation — computing $v$'s activations requires aggregating $u$'s activations.

The standard GNN-graph representation can be considered as a special case in the HAG representation with no intermediate aggregation nodes (i.e., $\mathcal{V}_A = \emptyset$). Our HAG abstraction is general and applicable to many existing GNN models. Table 1 shows how to use our abstraction to define existing GNNs, which can be further divided into two categories.

- **Set AGGREGATE**. Most GNNs assume the neighbors of a node have *no ordering*, and the aggregations are *associative* and *commutative* operations that are invariant to the order in which the aggregations are performed. Examples include GCN with summation aggregations and GraphSAGE-P with element-wise pooling aggregations (Table 1). Note that set aggregations in GNNs are designed to be order invariant and thus can be performed in a hierarchical fashion as we do in HAGs.
- **Sequential AGGREGATE**. Another class of GNNs require a specific ordering of a node's neighbors and the aggregations are not commutative. Examples include $N$-ary Tree-LSTM (Tai et al., 2015) and the LSTM variant of GraphSAGE (Hamilton et al., 2017). HAGs can be applied in the case of sequential aggregations as well. Rather than identifying common subsets of neighbors, we identify the common prefixes of the sequence of aggregated nodes, which can then be reused among nodes.

We further define two properties for the aggregation nodes $\mathcal{V}_A$. First, for each $v \in \mathcal{V}_A$, $\widehat{a}_v$ denotes its intermediate aggregation result, and $\widehat{\mathcal{N}}(v)$ denotes the in-neighbors of node $v$. To capture the aggregation hierarchy in a HAG, we use a recursive function to define $\widehat{a}_v$.

$$\widehat{a}_v = \text{AGGREGATE}\left( \left\{ \begin{array}{ll} h_u^{(k-1)} & u \in \mathcal{V} \\ \widehat{a}_u & u \in \mathcal{V}_A \end{array} \right| u \in \widehat{\mathcal{N}}(v) \right) \tag{1}$$

Second, $\widehat{\mathcal{C}}(v)$ denotes the set of input activations $h_u^{(k-1)}$ used to compute $\widehat{a}_v$ in the recursive procedure.

$$\widehat{a}_v = \text{AGGREGATE}(\{h_u^{(k-1)} | u \in \widehat{\mathcal{C}}(v)\}) \tag{2}$$

Intuitively, $\widehat{\mathcal{C}}(v)$ defines the coverage of node $v$ in a HAG. For the HAG example in Figure 1c, $\widehat{\mathcal{C}}(A) = \{B, C, D\}$ because $h_B^{(k-1)}$, $h_C^{(k-1)}$, and $h_D^{(k-1)}$ are used as inputs to compute $h_A^{(k)}$.

For a set AGGREGATE, $\widehat{\mathcal{C}}(\cdot)$ is an unordered set:

$$\widehat{\mathcal{C}}(v) = \bigcup_{u \in \widehat{\mathcal{N}}(v)} \left\{ \begin{array}{ll} u & u \in \mathcal{V} \\ \widehat{\mathcal{C}}(u) & u \in \mathcal{V}_A \end{array} \right. \tag{3}$$

For a sequential AGGREGATE, $\widehat{\mathcal{C}}(\cdot)$ is an ordered list:

$$\widehat{\mathcal{C}}(v) = \left( \widehat{\mathcal{C}}(u_1), ..., \widehat{\mathcal{C}}(u_m) \right) \tag{4}$$

where $u_1, ..., u_m$ are the ordered in-neighbors of $v$.

## 3.2 GNNs WITH HAGs

We extend the GNN abstraction to make it also applicable to HAGs, as shown in Algorithm 2. The extension does not require any modification to a GNN model, and the only difference is how to compute neighborhood aggregations (i.e., $a_v^{(k)}$) in each GNN layer. Before the original aggregations, we add one step that precomputes commonly used intermediate aggregation results to reduce redundant computation. In Algorithm 2, we first compute $\widehat{a}_v$ for all aggregation nodes $\mathcal{V}_A$ (line 3-4). This is performed recursively following the aggregation hierarchy of the HAG. We then compute the neighborhood aggregations $a_v^{(k)}$ (line 5-6) using the precomputed intermediate aggregations $\widehat{a}_v$.

**Algorithm 2** A GNN abstraction with HAGs. We exclude layer index superscripts in $\widehat{a}_v$ to denote that $\widehat{a}_v$ does not need to be memoized for back propagation, and its memory can be reused across all layers.

---

1: $h_v^{(0)} = x_v, \forall v \in \mathcal{V}$
2: **for** $k = 1$ to $K$ **do**
3:     **for** $v \in \mathcal{V}_A$ **do**
4:         compute $\widehat{a}_v$ using Equation (1)
5:     **for** $v \in \mathcal{V}$ **do**
6:         $a_v^{(k)} \leftarrow \text{AGGREGATE}(\{\widehat{a}_u | u \in \widehat{\mathcal{N}}_v\})$
7:         $h_v^{(k)} \leftarrow \text{UPDATE}(a_v^{(k)}, h_v^{(k-1)})$

---

**Memory overhead.** Although Algorithm 2 includes new intermediate variables $\widehat{a}_v$, the memory overhead for storing $\widehat{a}_v$ is negligible since $\widehat{a}_v$ is not used for back propagation and can be saved in a constant memory across all GNN layers. In the experiments, we show that HAGs can increase the training throughput by $2.8\times$, while maintaining the original model accuracy at the cost of 0.1% memory overhead to save $\widehat{a}_v$. Meanwhile, storing a HAG representation requires less memory than the original GNN-graph, as HAGs generally contain much fewer edges.

**Equivalence between GNN-graphs and HAGs.** We define a GNN-graph and a HAG to be *equivalent* if they produce the exact same outputs and gradients for a GNN model. Formally, a GNN-graph $\mathcal{G}$ and a HAG $\widehat{\mathcal{G}}$ are equivalent for a GNN model if (1) the GNN model outputs the same activations (i.e., $h_v^{(k)}$) at each GNN layer, and (2) the GNN model computes the same gradients for all trainable parameters in back propagation. Equivalent graphs guarantee the same predictive performance, and therefore can be used interchangeably for both GNN training and inference. Theorem 1 provides a simple yet efficient condition to check graph equivalence. We prove the theorem in the Appendix.

**Theorem 1.** *A GNN-graph with nodes $\mathcal{V}$ and a HAG with nodes $(\mathcal{V}, \mathcal{V}_A)$ are equivalent if and only if $\mathcal{N}(v) = \widehat{\mathcal{C}}(v)$ for all $v \in \mathcal{V}$, where $\mathcal{N}(v)$ is $v$'s neighbors in the input graph and $\widehat{\mathcal{C}}(\cdot)$ is defined in Equation 3 and 4.*

## 4 HAG SEARCH ALGORITHM

For a GNN model and an input GNN-graph, there exists a large space of equivalent HAGs with the same model accuracy but various runtime performance. Our goal is to explore the search space to discover a HAG with optimized runtime performance. First, we define a realistic cost function to quantitatively evaluate the runtime performance of different HAGs. Second, we introduce an efficient search algorithm that finds an optimized HAG with the following guarantees:

- For GNNs with sequential AGGREGATE, the HAG search algorithm can find *globally optimal* HAGs under the cost function.
- For GNNs with set AGGREGATE, finding an optimal HAG is NP-hard by a reduction from the NP-hard *maximum coverage problem* (see Appendix for the proof). The search algorithm finds at least a $(1 - 1/e)$-*approximation* of globally optimal HAGs based on the submodularity property.

### 4.1 COST FUNCTION

We introduce a cost function that estimates the runtime performance of a HAG by measuring the computation cost to perform one epoch of GNN training on the HAG.

The computation cost of a GNN model includes aggregating the neighbors of each node by calling AGGREGATE and updating the activations of each node via UPDATE, as shown in Algorithm 2. For a GNN model $\mathcal{M}$, we assume the cost of performing AGGREGATE on two elements is $\alpha_{\mathcal{M}}$, and the cost of computing an UPDATE is $\beta_{\mathcal{M}}$. In Algorithm 2, computing $\widehat{a}_v$ with $|\widehat{\mathcal{N}}_v|$ neighbors requires performing $(|\widehat{\mathcal{N}}_v| - 1)$ binary aggregations, whose cost is $\alpha_{\mathcal{M}} \times (|\widehat{\mathcal{N}}_v| - 1)$. Therefore, the total computation cost of training a GNN model $\mathcal{M}$ on a HAG $\widehat{\mathcal{G}}$ is

$$cost(\mathcal{M}, \widehat{\mathcal{G}}) = \sum_{v \in \mathcal{V} \cup \mathcal{V}_A} \alpha_{\mathcal{M}}(|\widehat{\mathcal{N}}_v| - 1) + \sum_{v \in \mathcal{V}} \beta_{\mathcal{M}} = \alpha_{\mathcal{M}}(|\widehat{\mathcal{E}}| - |\mathcal{V}_A|) + (\beta_{\mathcal{M}} - \alpha_{\mathcal{M}})|\mathcal{V}|$$

Since $|\mathcal{V}|$ is determined by the input graph, our goal is to minimize $(|\widehat{\mathcal{E}}| - |\widehat{\mathcal{V}}_A|)$.

### 4.2 SEARCH ALGORITHM

We present a HAG search algorithm that finds a globally optimal HAG for GNNs with sequential AGGREGATE and a $(1 - 1/e)$-approximation of globally optimal HAGs for GNNs with set AGGREGATE. In addition to an input GNN-graph and a GNN model, the algorithm also takes a hyper-parameter *capacity*, defining an upper limit on the number of intermediate aggregation nodes (i.e., $|\mathcal{V}_A|$).

Algorithm 3 shows the pseudocode of the HAG search algorithm. We start with an input GNN-graph, and iteratively insert aggregation nodes into the current HAG to merge highly redundant aggregations and remove unnecessary computation and data transfers.

In each iteration, we find a binary aggregation with the highest *redundancy* and insert a new aggregation node $w$ in $\mathcal{V}_A$ to represent the binary aggregation results (line 12-15). All nodes containing this binary aggregation can directly use the output of $w$ without recomputing the aggregation (line 16-18). The search algorithm iteratively reduces the computation cost of the HAG by eliminating the most redundant aggregation in each iteration. The redundancy scores are maintained in a *heap* structure.

---

**Algorithm 3** A HAG search algorithm to automatically find an equivalent HAG for a GNN-graph with optimized runtime performance. $\widehat{\mathcal{E}}$ and $\mathcal{V}_A$ are the set of edges and aggregation nodes in the HAG. REDUNDANCY$(v_1, v_2, \widehat{\mathcal{E}})$ calculates the number of nodes aggregating both $v_1$ and $v_2$. Recall that $\widehat{\mathcal{C}}(u)$ is an ordered list for sequential AGGREGATE (see Equation 4).

---

1: **Input:** A GNN-graph $\mathcal{G}$ and a GNN model $\mathcal{M}$.
2: **Output:** An equivalent HAG with optimized performance
3: **function** REDUNDANCY$(v_1, v_2, \widehat{\mathcal{E}})$
4:     **if** $\mathcal{M}$ has a set AGGREGATE **then**
5:         $\mathcal{R} = \{u | (v_1, u) \in \widehat{\mathcal{E}} \wedge (v_2, u) \in \widehat{\mathcal{E}}\}$
6:     **else**
7:         $\mathcal{R} = \{u | v_1 = \widehat{\mathcal{C}}(u)[1] \wedge v_2 = \widehat{\mathcal{C}}(u)[2]\}$
8:     **return** $|\mathcal{R}|$
9:
10: $\mathcal{V}_A \leftarrow \emptyset, \widehat{\mathcal{E}} \leftarrow \mathcal{E}$
11: **while** $|\mathcal{V}_A| <$ *capacity* **do**
12:     $(v_1, v_2) = \arg\max_{v_1, v_2}$ REDUNDANCY$(v_1, v_2, \widehat{\mathcal{E}})$
13:     **if** REDUNDANCY$(v_1, v_2, \widehat{\mathcal{E}}) > 1$ **then**
14:         $\mathcal{V}_A \leftarrow \mathcal{V}_A + \{w\}$                 $\triangleright$ where $w$ is a new node
15:         $\widehat{\mathcal{E}} \leftarrow \widehat{\mathcal{E}} + (v_1, w) + (v_2, w)$
16:         **for** $u \in \mathcal{V}$ **do**
17:             **if** $(v_1, u) \in \widehat{\mathcal{E}} \wedge (v_2, u) \in \widehat{\mathcal{E}}$ **then**
18:                 $\widehat{\mathcal{E}} \leftarrow \widehat{\mathcal{E}} - (v_1, u) - (v_2, u) + (w, u)$
19: **return** $(\mathcal{V}_A \cup \mathcal{V}, \widehat{\mathcal{E}})$

---

For a GNN model with a sequential AGGREGATE, Theorem 2 shows that our search algorithm finds an equivalent HAG with globally optimal computation cost. We prove the theorem in Appendix.

**Theorem 2.** *For any GNN-graph $\mathcal{G} = (\mathcal{V}, \mathcal{E})$ and any GNN model $\mathcal{M}$ with a sequential AGGREGATE, Algorithm 3 returns an equivalent HAG with globally minimum cost as long as capacity $\geq |\mathcal{E}|$.*

For a GNN model with a set AGGREGATE, Theorem 3 shows that our search algorithm finds a HAG that is at least a $(1 - 1/e)$-approximation of the globally optimal HAGs (see proof in Appendix).

**Theorem 3.** *For any GNN-graph $\mathcal{G}$ and GNN model $\mathcal{M}$ with a set AGGREGATE, Algorithm 3 gives a $(1 - 1/e)$-approximation of globally optimal HAGs under the cost function. Formally, let $\widehat{\mathcal{G}}$ be the HAG returned by Algorithm 3, and $\widehat{\mathcal{G}}_o$ is a globally optimal HAG under the capacity constraint,*

$$cost(\mathcal{M}, \widehat{\mathcal{G}}) \leq \frac{1}{e} cost(\mathcal{M}, \mathcal{G}) + \frac{e-1}{e} cost(\mathcal{M}, \widehat{\mathcal{G}}_o)$$

**Time and space complexity.** Our HAG algorithm achieves low theoretical complexity and has negligible runtime overhead. In particular, the overall time complexity of Algorithm 3 is $O(capacity \times |\mathcal{V}| + |\mathcal{E}| \times \log |\mathcal{V}|)$, and the space complexity is $O(capaccity \times |\mathcal{V}| + |\mathcal{E}|)$ (see Appendix for the proof). One key optimization is a heap data structure for maintaining the redundancy scores of the highest $O(|V|)$ node pairs. Finding the most redundant node pair over the entire graph thus only takes $O(1)$ time, and updating the redundancy scores (i.e., line 15 and 18) each takes $O(\log |V|)$ time.

## 5 EXPERIMENTS

The HAG representation maintains the predictive performance of GNNs but has much better runtime performance. This section evaluates the runtime performance of HAGs on five real-world graph datasets. We evaluate HAGs along three dimensions: (a) end-to-end training and inference performance; (b) number of aggregations; and (c) size of data transfers.

### 5.1 IMPLEMENTATION

Existing deep learning frameworks such as TensorFlow and PyTorch are designed for spatial data structures (e.g., images and text), and have limited support for irregular data structures such as graphs.

As a result, GNN models in existing frameworks translate graph structures to sparse adjacent matrices and use matrix operations to perform GNN training.

We implemented the following operations in TensorFlow r1.14 to support GNN training with HAGs. First, `graph_to_hag` automatically transforms an input GNN-graph to an equivalent HAG with optimized performance. Second, `hag_aggregate` takes a HAG and nodes' activations as inputs, and computes the aggregated activations of all nodes. Finally, `hag_aggregate_grad` computes the gradients of `hag_aggregate` for back propagation. Our implementation minimizes changes to existing GNN programs: a GNN application can directly use all HAG optimizations by only modifying a few lines of code.

## 5.2 EXPERIMENTAL SETUP

**Datasets.** Table 2 summarizes the public datasets used in our experiments. BZR is a chemical compound dataset, where each node is an atom and an edge is a chemical bond between two atoms (Kriege and Mutzel, 2012). PPI contains a number of protein-protein interaction graphs, each of which corresponds to a different human tissue (Zitnik and Leskovec, 2017). REDDIT is an online discussion forum dataset, with each node being a Reddit post and each edge being commenting relations. For both

Table 2: Datasets used in the experiments.

| Name | # Nodes | # Edges |
|------|---------|---------|
| Node Classification | | |
| BZR | 6,519 | 137,734 |
| PPI | 56,944 | 1,612,348 |
| REDDIT | 232,965 | 114,615,892 |
| Graph Classification | | |
| IMDB | 19,502 | 197,806 |
| COLLAB | 372,474 | 12,288,900 |

PPI and REDDIT, we directly use prepossessed data from Hamilton et al. (2017). IMDB and COLLAB are two collaboration datasets for graph classification (Yanardag and Vishwanathan, 2015). IMDB is a movie collaboration dataset, with each node representing an actor/actress, while COLLAB is a scientific collaboration dataset, with each node representing a researcher.

All experiments were performed running TensorFlow r1.14 on NVIDIA Tesla V100 GPUs. Following previous work (Kipf and Welling, 2016; Hamilton et al., 2017), each GNN model has two GNN layers and one SoftMax layer. For graph classification datasets, each GNN model also includes a mean-pooling layer to gather graph-level activations. For all experiments, we set the maximum *capacity* of $|\mathcal{V}_A|$ in a HAG to be $|\mathcal{V}|/4$, which achieves high performance on real-world graphs.

## 5.3 END-TO-END PERFORMANCE

We first measure the per-epoch training time and inference latency to run a 2-layer GCN model on different graph datasets. We follow previous work (Hamilton et al., 2017; Kriege and Mutzel, 2012; Yanardag and Vishwanathan, 2015) to split the datasets into training/validation/testing sets, and use the testing sets to measure the average inference latency.

Figure 2 compares the per-epoch training time and inference latency between GNN-graphs and HAGs. Compared to GNN-graphs, HAGs can improve the training and inference performance by up to $2.8\times$ and $2.9\times$, respectively, while maintaining the same network accuracy. We note this improvement is achieved completely automatically, and computing a HAG is inexpensive. Thus, because the improvement is essentially for free, we believe there is no reason not to use HAGs in preference to GNN-graphs.

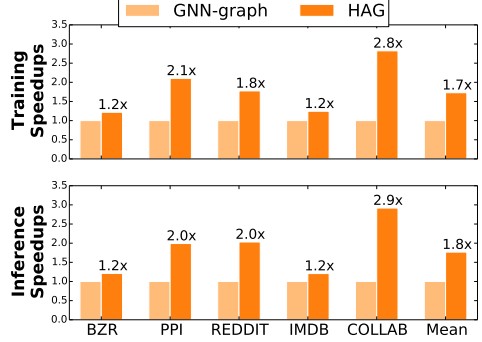

Figure 2: End-to-end performance comparison between GNN-graphs and HAGs. We measure the per-epoch training time and inference latency on a 2-layer GCN model with 16 hidden dimensions in each layer. The performance numbers are normalized by the GNN-graph numbers.

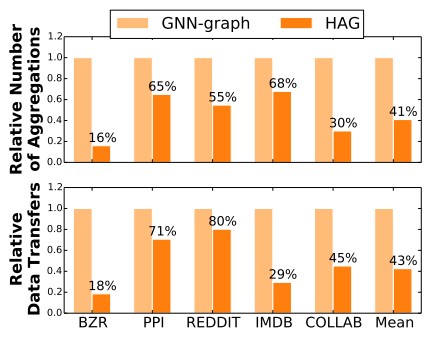 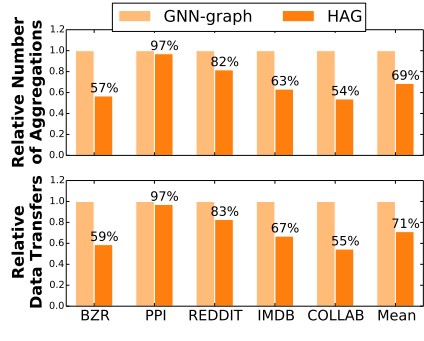

(a) Set Aggregations.                    (b) Sequential Aggregations.

Figure 3: Comparing the number of aggregations and amount of data transfers between GPU threads to perform aggregations (lower is better). The y-axes are normalized by GNN-graphs, and the last column in each figure is the geometry mean over all datasets.

## 5.4 AGGREGATION PERFORMANCE

We further compare the aggregation performance of GNN-graphs and HAGs on the following two metrics: (1) the number of binary aggregations performed in each GNN layer; and (2) the size of data transfers between GPU threads to perform the aggregations. Note that aggregating a neighbor's activations requires transferring the activations from GPU global memory to a thread's local memory.

Figure 3 shows the comparison results. For GNNs with set aggregations, HAGs reduce the number of aggregations by 1.5-6.3× and the size of data transfers by 1.3-5.6×. For GNNs with sequential aggregations, HAGs reduce aggregations and data transfers by up to 1.8× and 1.9×, respectively.

Although the search algorithm finds a globally optimal HAG for sequential aggregations (Theorem 2) and a $(1 - 1/e)$-approximation of globally optimal HAGs for set aggregations (Theorem 3), we observe the performance improvement is more significant for set aggregations. Optimality for HAGs with set aggregation involves more potential redundancy compared to sequential aggregations, due to permutation invariance of set aggregation. Thus higher performance can be achieved with HAGs for set aggregations, though optimal solutions are more difficult to compute.

It is also worth noting that the HAG search algorithm can find highly optimized HAGs even on very sparse graphs. For example, on the COLLAB dataset with a graph density of 0.01%, our algorithm reduces the number of aggregations and data transfers by 3.3× and 2.2×, respectively.

## 5.5 HAG SEARCH ALGORITHM

We evaluate the performance of the HAG search algorithm. Recall that the search algorithm uses a hyper-parameter *capacity* to control the number of aggregation nodes in a HAG. A larger *capacity* allows the algorithm to eliminate more redundant aggregations and achieves lower cost.

Figure 4 shows the end-to-end GCN training time on the COLLAB dataset using HAGs with different capacities. A larger value of capacity can consistently improve the training performance, which indicates that the cost function is an appropriate metric to evaluate and compare the performance of different HAGs. By gradually increasing the capacity, the search algorithm eventually finds a HAG with ~100K aggregation nodes, which consume 6MB of memory (0.1% memory overhead) while improving the training performance by 2.8×. In addition, the HAG search time is negligible compared to the end-to-end training time.

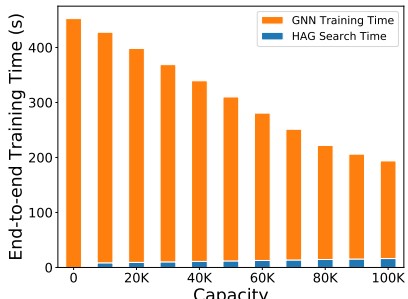

Figure 4: End-to-end GCN training time on the COLLAB dataset using HAGs with different capacities. We train GCN for a maximum of 350 epochs by following prior work (Kipf and Welling, 2016).

**Limitation.** One limitation of the HAG graph representation is that it does not apply to attention-based GNN models, such as GAT (Veličković et al., 2017), which uses the attention scores (Vaswani et al., 2017) between nodes to guide activation updates and results in limited redundancy that can be optimized by HAG.

**Conclusion.** We introduce HAG, a new graph representation to eliminate redundancy in GNNs. We propose a cost function to estimate the performance of different HAGs and use a search algorithm to find optimized HAGs. We show that HAGs outperform existing GNN-graphs by improving the end-to-end training performance and reducing the aggregations and data transfers in GNN training.

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

## A   PROOF OF THEOREM 1

*Proof.* It is sufficient to prove that if $\mathcal{N}(v) = \widehat{\mathcal{C}}(v)$ for all $v \in \mathcal{V}$, then the GNN-graph $\mathcal{G}$ and the HAG $\widehat{\mathcal{G}}$ generate the same outputs (i.e., $h_v^{(k)}$) for every GNN layer.

We prove this by induction. Assume a GNN-graph $\mathcal{G}$ and a HAG $\widehat{\mathcal{G}}$ generate the same outputs for the $(k\text{-}1)$-th layer, we prove the two graphs produce the same outputs for the $k$-th GNN layer.

In Algorithm 2, $\widehat{a}_v$ is the aggregation results of node $v$, which is defined as

$$\begin{aligned} \widehat{a}_v &= \text{AGGREGATE}(h_u^{(k-1)}|u \in \widehat{\mathcal{C}}(v)) \\ &= \text{AGGREGATE}(h_u^{(k-1)}|u \in \mathcal{N}(v)) \end{aligned}$$

This proves that Algorithm 1 and Algorithm 2 compute the same $a_v^{(k)}$. In addition, both algorithms use the same UPDATE function that takes $a_v^{(k)}$ and $h_v^{(k-1)}$ as inputs and computes $h_v^{(k)}$, which applies that the two algorithms compute the same $h_v^{(k)}$. □

## B   PROOF OF THEOREM 2

*Proof.* Sequential aggregations require a specific ordering of a node's neighbors. Let $\mathcal{N}_v$ denote the ordered list of node $v$'s neighbors and $\mathcal{L}_v^{(i)}$ denote a list of the first $i$ elements in $\mathcal{N}_v$:

$$\mathcal{L}_v^{(i)} = \big(\mathcal{N}_v(1), \mathcal{N}_v(2), ..., \mathcal{N}_v(i)\big)$$

where $\mathcal{N}_v(i)$ is the $i$-th neighbor of node $v$.

$\mathcal{L}_v^{(i)}$ represents a necessary intermediate aggregation step for computing $a_v^{(k)}$ (since sequential aggregations are not commutative), and therefore any HAG must compute $\mathcal{L}_v^{(i)}$ as an intermediate aggregation. Counting the number of distinct $\mathcal{L}_v^{(i)}$ (where $v \in \mathcal{V}$ and $1 < i \le |\mathcal{N}_v|$) provides a lower bound on the number of aggregations any equivalent HAG must perform. Assuming $\widehat{\mathcal{G}}_o$ is a globally optimal HAG under the cost model, we have:

$$cost(\mathcal{M}, \widehat{\mathcal{G}}_o) \ge \alpha_{\mathcal{M}} \times lb + (\beta_{\mathcal{M}} - \alpha_{\mathcal{M}})|\mathcal{V}|$$

where $lb$ is the number of distinct $\mathcal{L}_v^{(i)}$ that must be computed by any equivalent HAG.

Assuming $\widehat{\mathcal{G}}$ is the output HAG of Algorithm 3, we prove that $cost(\mathcal{M}, \widehat{\mathcal{G}}) = cost(\mathcal{M}, \widehat{\mathcal{G}}_o)$ by using contradiction. In the case $cost(\mathcal{M}, \widehat{\mathcal{G}}) > cost(\mathcal{M}, \widehat{\mathcal{G}}_o)$, $\widehat{\mathcal{G}}$ must perform more than $lb$ aggregations.

**Case 1.** One possible case is that $\widehat{\mathcal{G}}$ computes at least one aggregation that is not a prefix of any $\mathcal{N}_v$, indicating that $\widehat{\mathcal{G}}$ performs some useless aggregations, which contradicts with the fact that all intermediate aggregations added to $\widehat{\mathcal{G}}$ must be used at least once.

**Case 2.** The other possible case is that $\widehat{\mathcal{G}}$ computes the aggregation of some $\mathcal{L}_v^{(i)}$ multiple times. However, in Algorithm 3, each iteration reduces the number of aggregations by at least 1, and there are $|\mathcal{E}|$ aggregations initially. This implies there cannot be redundant aggregations after $|\mathcal{E}|$ iterations, which contradicts with the precondition of Case 2. □

## C   PROOF OF THEOREM 3

*Proof.* The idea of the proof is to build a *monotone submodular function* Cormen et al. (2009) based on the cost model.

For any GNN-graph $\mathcal{G}$ and an equivalent $\widehat{\mathcal{G}}$, we define

$$
\begin{aligned}
f(\widehat{\mathcal{G}}) &= cost(\mathcal{M}, \mathcal{G}) - cost(\mathcal{M}, \widehat{\mathcal{G}}) & (5) \\
&= \alpha_{\mathcal{M}}(|\mathcal{E}| - |\widehat{\mathcal{E}}| + |\mathcal{V}_A|) & (6)
\end{aligned}
$$

where $\mathcal{V}_A$ is the set of aggregation nodes in $\widehat{\mathcal{G}}$, and $\mathcal{E}$ and $\widehat{\mathcal{E}}$ are the set of edges in $\mathcal{G}$ and $\widehat{\mathcal{G}}$, respectively. $f(\widehat{\mathcal{G}})$ measures the number of aggregations that can be saved by using $\widehat{\mathcal{G}}$ for GNN training.

We begin by defining the subset relations between different HAGs. For two HAGs $\widehat{\mathcal{G}}$ and $\widehat{\mathcal{G}}'$, we define $\widehat{\mathcal{G}} \subseteq \widehat{\mathcal{G}}'$ iff $\mathcal{V}_A$ is a subset of $\mathcal{V}'_A$, where $\mathcal{V}_A$ and $\mathcal{V}'_A$ are the aggregation nodes in $\widehat{\mathcal{G}}$ and $\widehat{\mathcal{G}}'$, respectively.

**Prove that $f(\widehat{\mathcal{G}})$ is monotone.** We show that for all $\widehat{\mathcal{G}} \subseteq \widehat{\mathcal{G}}'$, $f(\widehat{\mathcal{G}}) \leq f(\widehat{\mathcal{G}}')$. This is true since $\widehat{\mathcal{G}} \subseteq \widehat{\mathcal{G}}'$ indicates that $\widehat{\mathcal{G}}'$ contains all aggregation nodes in $\widehat{\mathcal{G}}$, which applies that $\widehat{\mathcal{G}}'$ can at least save the same number of aggregations as $\widehat{\mathcal{G}}$.

**Prove that $f(\widehat{\mathcal{G}})$ is submodular.** We show that for all $\widehat{\mathcal{G}} \subseteq \widehat{\mathcal{G}}'$ and any aggregation node $n$, $f(\widehat{\mathcal{G}} + \{n\}) - f(\widehat{\mathcal{G}}) \geq f(\widehat{\mathcal{G}}' + \{n\}) - f(\widehat{\mathcal{G}}')$. This inequality holds because $f(\widehat{\mathcal{G}} + \{n\}) - f(\widehat{\mathcal{G}})$ measures the number of aggregations we can further save by adding aggregation $n$ to the existing HAG, which monotonically decreases as we add more aggregation nodes to the HAG.

Let $\widehat{\mathcal{G}}^{(i)}$ denote the result HAG after the $i$-th iteration of Algorithm 3. $\widehat{\mathcal{G}}^{(i)}$ includes exactly $i$ aggregation nodes. Let $\widehat{\mathcal{G}}_o$ denote the optimal HAG under the cost model with $k$ aggregation nodes. We claim via induction that for $0 \leq i \leq k$,

$$
f(\widehat{\mathcal{G}}_o) - f(\widehat{\mathcal{G}}^{(i)}) \leq (1 - 1/k)^i f(\widehat{\mathcal{G}}_o) \tag{7}
$$

The base case is trivially true. In the $i$-th step, Algorithm 3 selects an aggregation node $a_i$ by maximizing the marginal gain $f(\widehat{\mathcal{G}}^{(i)} + a_i) - f(\widehat{\mathcal{G}}^{(i)})$. Observe that the remaining aggregation nodes includes $\widehat{\mathcal{G}}_o \setminus \widehat{\mathcal{G}}^{(i-1)}$, a set of at most $k$ elements. The submodularity applies that

$$
f(\widehat{\mathcal{G}}_o) - f(\widehat{\mathcal{G}}^{(i-1)}) \leq \sum_{a \in \widehat{\mathcal{G}}_o \setminus \widehat{\mathcal{G}}^{(i-1)}} \left( f(\widehat{\mathcal{G}}^{(i)} + a) - f(\widehat{\mathcal{G}}^{(i)}) \right)
$$

and this implies that the aggregation node $a_i$ has marginal value

$$
\begin{aligned}
& f(\widehat{\mathcal{G}}^{(i-1)} + a_i) - f(\widehat{\mathcal{G}}^{(i-1)}) \\
\geq\; & \frac{1}{|\widehat{\mathcal{G}}_o \setminus \widehat{\mathcal{G}}^{(i-1)}|} \sum_{a \in \widehat{\mathcal{G}}_o \setminus \widehat{\mathcal{G}}^{(i-1)}} \left( f(\widehat{\mathcal{G}}^{(i)} + a) - f(\widehat{\mathcal{G}}^{(i)}) \right) \\
\geq\; & \frac{1}{k} \left( f(\widehat{\mathcal{G}}_o) - f(\widehat{\mathcal{G}}^{(i-1)}) \right)
\end{aligned}
$$

Assuming that Inequality 7 holds for $\widehat{\mathcal{G}}^{(i-1)}$, we have

$$
\begin{aligned}
f(\widehat{\mathcal{G}}_o) - f(\widehat{\mathcal{G}}^{(i)}) &= f(\widehat{\mathcal{G}}_o) - f(\widehat{\mathcal{G}}^{(i-1)}) - \left( f(\widehat{\mathcal{G}}^{(i)}) - f(\widehat{\mathcal{G}}^{(i-1)}) \right) \\
&\leq f(\widehat{\mathcal{G}}_o) - f(\widehat{\mathcal{G}}^{(i-1)} - \frac{1}{k}(f(\widehat{\mathcal{G}}_o) - f(\widehat{\mathcal{G}}^{(i-1)})) \\
&= (1 - 1/k)(f(\widehat{\mathcal{G}}_o) - f(\widehat{\mathcal{G}}^{(i-1)})) \\
&\leq (1 - 1/k)^i f(\widehat{\mathcal{G}}_o)
\end{aligned}
$$

which proves Inequality 7. Therefore, we have

$$
f(\widehat{\mathcal{G}}_o) - f(\widehat{\mathcal{G}}^{(k)}) \leq (1 - 1/k)^k f(\widehat{\mathcal{G}}_o) \leq e^{-1} f(\widehat{\mathcal{G}}_o)
$$

By taking in the definition of $f(\cdot)$, we have

$$
cost(\mathcal{M}, \widehat{\mathcal{G}}) \leq \frac{1}{e} cost(\mathcal{M}, \mathcal{G}) + \frac{e-1}{e} cost(\mathcal{M}, \widehat{\mathcal{G}}_o)
$$

$\square$

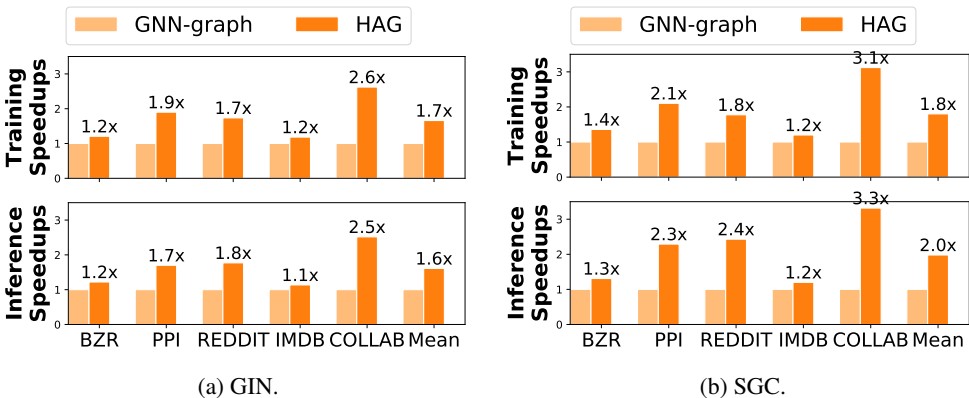

Figure 5: End-to-end performance comparison between GNN-graphs and HAGs. We measure the per-epoch training time and inference latency on a 2-layer GIN and a 2-layer SGC model. Both models have 16 hidden dimensions in each layer. The performance numbers are normalized by the GNN-graph numbers (higher is better).

## D  COMPLEXITY OF THE HAG SEARCH ALGORITHM

**Theorem 4.** *The overall time complexity of Algorithm 3 is $O(capacity \times |\mathcal{V}| + |\mathcal{E}| \times \log |\mathcal{V}|)$, where capacity is the upper bound on the number of aggregation nodes in a HAG.*

*Proof.* We use a *heap* to maintain the redundancy score of each potential node pair and only update the heap when we add and remove edges in $\widehat{\mathcal{E}}$. Since the depth of the heap is at most $O(\log |\mathcal{V}|)$ [1], querying the most redundant binary aggregation and modifying $\widehat{\mathcal{E}}$ each takes $O(\log |\mathcal{V}|)$ time.

First, we calculate the number of queries and updates to the heap structure:

- The algorithm iteratively pull the most redundant binary aggregation from the heap and add it to $\mathcal{V}_A$. Since the number of vertices in $\mathcal{V}_A$ is smaller than *capacity*, the total number of queries is $O(capacity)$.

- The algorithm inserts two new edges into $\widehat{\mathcal{E}}$ in line 16 and removes one edge from $\widehat{\mathcal{E}}$ in line 19. Since line 16 can be invoked at most $O(capacity)$ times, the total number of invocations to line 19 is $O(|\mathcal{E}| + 2 \times capacity)$. Therefore, the overall number of updates is $O(|\mathcal{E}| + capacity)$.

Second, the enumeration over all vertices in $\mathcal{V}$ (line 17) involves time complexity of $O(capacity \times |\mathcal{V}|)$. Therefore, the overall time complexity of Algorithm 3 is

$$O\big(capacity \times |\mathcal{V}| + (|\mathcal{E}| + capacity) \times \log |\mathcal{V}|\big)$$
$$= O(capacity \times |\mathcal{V}| + |\mathcal{E}| \times \log |\mathcal{V}|)$$

$\square$

## E  PERFORMANCE EVALUATION

We further evaluate the effectiveness of the HAG optimizations on two additional GNN models: GIN Xu et al. (2019) and SGC Wu et al. (2019). The experimental setup is described in Section 5.2. We measure the per-epoch training time and inference latency for GIN and SGC. We follow previous work Xu et al. (2019); Wu et al. (2019) to split the datasets into training/validation/testing sets, and use the testing sets to measure the average inference latency.

Figure 5 compares the performance between GNN-graphs and HAGs. Compared to GNN-graphs, HAGs improve the training performance by up to 2.6× and 3.1× on GIN and SGC, while maintaining the same network accuracy. This shows that the HAG optimization is general and applicable to various GNN models.

---

[1] This is because there can be at most $O(|\mathcal{V}|^2)$ node pairs.

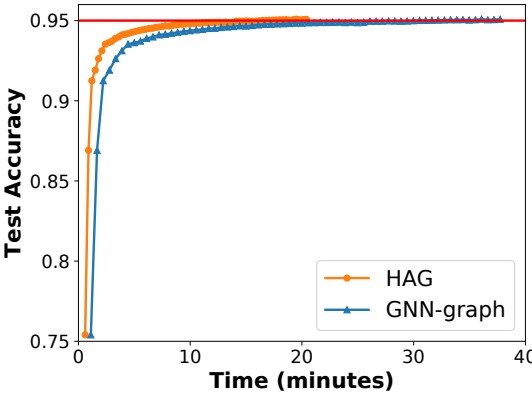

Figure 6: Time-to-accuracy comparison between HAG and GNN-graph for training a 2-layer GCN model on the Reddit dataset.

## F    TIME-TO-ACCURACY EVALUATION

We compare the time-to-accuracy performance between HAG and GNN-graph. We train a 2-layer GCN model (with 64 hidden dimensions in each layer) on the Reddit dataset until the test accuracy exceeds 95%. We follow previous work Kipf and Welling (2016) to set all hyper-parameters and split the dataset.

Figure 6 shows the results. Each dot indicates the training time and test accuracy of each epoch. Training the GCN model using the HAG representation achieves the same training and test accuracy at the end of each epoch. This is because the HAG optimization maintains the same results as the original GNN-graph representation and therefore preserves the model accuracy. It takes 55 training epochs to achieve a test accuracy of 95% for both HAG and GNN-graph, and HAG improves the end-to-end training time by $1.8\times$.

