# OpenReview forum: "Redundancy-Free Computation Graphs for Graph Neural Networks"
_ICLR.cc/2020/Conference — Reject_

### Official Review · AnonReviewer2 · 2019-10-23
**Official Blind Review #32**

**Rating:** 6

**Review:**

In this paper, authors propose a way to speed up the computation of GNN. More specifically, the hierarchically aggregate computation graphs are proposed to aggregate the intermediate node and utilize this to speed up a GNN computation. Authors proof that the computation based on HAGs are equivalent to the vanilla computation of GNN (Theorem1). Moreover, for sequential aggregation, it can find a HAG that is at least (1-1/e)-approximation of the globally optimal HAGs (Theorem 3). These theoretical results are nice. Through experiments, the authors demonstrate that the proposed method can get faster computation than vanilla algorithms.

The paper is clearly written and easy to follow. However, there are some missing piece needed to be addressed.
I put 6 (weak accept), since we cannot put 5. However, current my intention about the score is slightly above 5.

Detailed comments:
1. Experiments are only done for computational time comparison. In particular, for the sequential one, prediction accuracy can be changed due to the aggregation algorithm. Thus, it needs to report the prediction accuracy.

2. In GraphSAGE, what is the sampling rate? It would be nice to have the trade-off between the sampling rate and the speedup. I guess if we sample small number of points in GraphSAGE, the performance can be degraded. In contrast, the proposed algorithm can get similar performance with larger sampling rate? Related to the question 1, the performance comparison is needed.

3. Equations are used without not explaining the meaning. For instance AGGREGATE function (1), there is no definition how to aggregate.


**Experience Assessment:**

I have published one or two papers in this area.

**Review Assessment: Checking Correctness Of Derivations And Theory:**

I assessed the sensibility of the derivations and theory.

**Review Assessment: Checking Correctness Of Experiments:**

I assessed the sensibility of the experiments.

**Review Assessment: Thoroughness In Paper Reading:**

I read the paper at least twice and used my best judgement in assessing the paper.

---

> ### Author Response · Authors · 2019-11-15
> **Response to Review #2**
>
> We thank the reviewer for a thorough review and valuable questions. The reviewer has concerns about (1) how HAG affects model accuracy and (2) how sampling rate affects the runtime performance. We believe there are a few important misunderstandings, which we address in detail in our response below. We have also updated the paper to further clarify and emphasize these points.
>
> ### Need to report the model accuracy
> The HAG graph representation achieves the same training and test results as the original GNN-graph representation for each GNN model, even though GNN models with different aggregation methods (i.e., set v.s. sequential) may obtain different accuracy--- this issue is orthogonal to the HAG optimizations.
>
> Theorem 1 in the paper proves the equivalence of HAG and GNN-graph representations for both training and inference. This means that HAG performs exactly the same computations as the traditional model training/inference but in a non-redundant way. This means, that HAG obtains exactly the same model as traditional training (but HAG is much faster).
>
> To address reviewer’s comment we have added an experiment to evaluate the training effectiveness of HAG (Figure 6 on page 13), which compares the time-to-accuracy performance between original GNN-graph representation and HAG, and show that HAG can reduce the training time by 1.8x while obtaining the same model accuracy.
>
> ### What is the trade-off between the sampling rate and the speedup
> We think there is a misunderstanding here. The reviewer seems to assume that HAG is designed for mini-batch training, probably because we use GraphSAGE as an example to demonstrate different aggregation functions in Table 1. In fact, HAG is designed for full-batch training, which is also the training method used in most existing GNN models, including GCN (Kipf & Welling, 2016), GIN (Xu et al., 2019), and SGC (Wu et al., 2019). We will fix this confusion by emphasizing the full-batch training setting in the introduction and use other GNN models (with full-batch training) as examples in Table 1.
>
> ### Equations are used without explaining the meaning (e.g., AGGREGATE in Equation (1))
> We apologize for the missing explanation in the equations. The AGGREGATE in Equation (1) can be arbitrary associative and commutative operations performed on a set (i.e., invariant to the order in which the aggregations are performed). We have updated the paper to clarify this.

---

> > ### Comment · AnonReviewer2 · 2019-11-15
> > **Thank you for the response**
> >
> >
> > Thank you for the clarification for GraphSAGE. I understand.
> >
> > >The HAG graph representation achieves the same training and test results as the original GNN-graph representation for each GNN model, even though GNN models with different aggregation methods (i.e., set v.s. sequential) may obtain different accuracy--- this issue is orthogonal to the HAG optimizations.
> > >Theorem 1 in the paper proves the equivalence of HAG and GNN-graph representations for both training and inference. This means that HAG performs exactly the same computations as the traditional model training/inference but in a non-redundant way. This means, that HAG obtains exactly the same model as traditional training (but HAG is much faster).
> > >To address reviewer’s comment we have added an experiment to evaluate the training effectiveness of HAG (Figure 6 on page 13), which compares the time-to-accuracy performance between original GNN-graph representation and HAG, and show that HAG can reduce the training time by 1.8x while obtaining the same model accuracy.
> >
> > Thanks for the comment.
> > I am completely fine for the exact case (i.e., Theorem 1). This case the performance should be the same.
> >
> > So, Figure 6 is based on Theorem1 or Theorem 2? This part is still not clear.
> > If this is exact one (based on Theorem 1),  I still believe the accuracy comparison of sequential one is necessary, since we can make the method extremely fast with very poor accuracy.

---

> > > ### Author Response · Authors · 2019-11-15
> > > **Thank you for the quick response**
> > >
> > > It seems there is a slight misunderstanding in Theorem 1 and 2.
> > >
> > > Theorem 1 shows that HAG always maintains the exact same model accuracy as GNN-graph for both categories of GNN models (i.e., set and sequential AGGREGATE). In fact, training a GNN model on HAG produces the exact same activations/gradients/weights as traditional training on GNN-graph in each epoch.
> > >
> > > Theorem 2 and 3 describe the runtime performance (i.e., the execution time to train an epoch) of the HAGs discovered by our search algorithm, since there exist numerous HAGs functionally equivalent to the original GNN-graph. In particular, Theorem 2 proves that, for GNN models with sequential AGGREGATE, the search algorithm finds a HAG with globally optimal runtime performance (i.e., minimal execution time to train an epoch). Theorem 3 proves a similar bound for GNN models with set AGGREGATE.
> > >
> > > Figure 6 compares the training time of HAG and GNN-graph for GCN, which uses set AGGREGATE. We are happy to also include a time-to-accuracy comparison for a GNN model with sequential AGGREGATE in the final paper. Note that Theorem 1 proves that HAG maintains the same model accuracy as the original GNN-graph by design.

---

### Official Review · AnonReviewer3 · 2019-10-24
**Official Blind Review #3**

**Rating:** 3

**Review:**

This paper aims to propose a speeding-up strategy to reduce the training time for existing GNN models by reducing the redundant neighbor pairs. The idea is simple and clear. The paper is well-written. However, major concerns are:

1. This strategy is only for equal contribution models (e.g., GCN, GraphSAGE), not for methods which consider distinct contribution weights for individual node neighbor (e.g. GAT). However, in my opinion, for one target node, different neighbors should be assigned with different contributions instead of equal contributions.

2. What kind of graphs can the proposed model be applied to? This paper seems to only consider unweighted undirected graphs. How about directed and weighted graphs? Even for an unweighted undirected graph, the symmetric information may also be redundant for further elimination. Then can HAG reduce this symmetric redundancy?

3. There is no effectiveness evaluation comparing the original GNN models with the versions with HAG. The authors claim that, with the HAG process, the efficiency could be improved without losing accuracy. But there are no experimental results verifying that the effectiveness of the HAG-versions which could obtain comparable performance with the original GNN models for some downstream applications (e.g., node classification).

--------------------------------------------------Update------------------------------------------------
Thanks very much for the authors' feedback. The revised version has clarified some of my concerns. However, the equal-contribution (in Comment 1) is still a big one that the authors should pay attention. I increase my score to 3.


**Experience Assessment:**

I have published one or two papers in this area.

**Review Assessment: Checking Correctness Of Derivations And Theory:**

I assessed the sensibility of the derivations and theory.

**Review Assessment: Checking Correctness Of Experiments:**

I carefully checked the experiments.

**Review Assessment: Thoroughness In Paper Reading:**

I read the paper at least twice and used my best judgement in assessing the paper.

---

> ### Author Response · Authors · 2019-11-15
> **Response to Review #3**
>
> We thank the reviewer for providing a thorough review and asking valuable questions. The reviewer raises several concerns about the broad applicability of HAGs. As we argue below, HAG is broadly applicable to the majority of GNN models, it can support directed and weighted graphs, and it provides significant speedups with no loss in model performance (both at training as well as prediction time). We have revised the paper to further clarify and emphasize all these points.
>
> ###  HAG does not support GAT
> We thank the reviewer for making this point. This is correct and HAG optimization does not apply to graph attention networks (GAT). The HAG graph representation is designed for GNNs with a neighborhood aggregation scheme (formally defined in Algorithm 1), and is applicable to most existing GNN models, including GraphSAGE, PinSAGE, GCN, GIN, SGC, DCNN, DGCNN, and many others. Thus, it is reasonable to conclude that HAG applies to a significant majority of GNN models. Because such GNN-graphs are processed individually in these GNN models, they contain significant redundant computation (up to 84%), and HAG can reduce the overall computation by 6.3x, while provably preserving the original model accuracy.
> We do appreciate the comment, and we have added a paragraph at the end of page 8 to discuss this limitation of HAG in the revised paper.
>
> ### Can HAG support directed and weighted graph?
> HAG can support directed and/or weighted graphs as long as the GNN models can be abstracted as in Algorithm 1. In particular, HAG can support directed graphs by changing N(v) in Algorithm 1 to be the set of incoming-neighbors of node v (instead of the set of all neighbors). For weighted graphs, HAG can incorporate edge weights in neighborhood aggregation by updating the AGGREGATE function in Algorithm 1 to consider edge weights. For weighted graphs, rather than identifying common subsets of neighbors, HAG identifies common neighbors with shared edge weights as redundant computation. The fact that existing GNN models in the literature do not consider edge weights and are designed for undirected graphs makes it hard to find a realistic benchmark to evaluate the performance of HAG on directed and weighted graphs.
>
> To address the reviewer’s point we have added a discussion on potential extensions of our HAG algorithm to directed and weighted graphs in the revised paper.
>
> ### The training effectiveness of HAG is questionable
> There is a slight misunderstanding here. HAG eliminates redundancy in GNN training while exactly maintaining the original computation (proved in Theorem 1), therefore it is guaranteed to preserve the original model accuracy by design. However, to address reviewer’s comment we have added an experiment to evaluate the training effectiveness of HAG (Figure 6 on page 13), which compares the time-to-accuracy performance between original GNN-graph representation and HAG, and shows that HAG can reduce the end-to-end training time by 1.8x while obtaining the same model accuracy. Thus, HAG leads to significant faster training time with no loss in model performance.
>
> We have further clarified this in the main paper, because we see it as one of the important benefits of HAG that it maintains original model performance (by performing exactly the same computations), while leading to significant speed-ups.

---

### Official Review · AnonReviewer4 · 2019-11-06
**Official Blind Review #4**

**Rating:** 6

**Review:**

This paper proposes a new graph Hierarchy representation named HAG. The HAG aiming at eliminating the redundancy during the aggregation stage In Graph Convolution networks. This strategy can speed up the training and inference time while keeping the GNN output unchanged, which means it can get the same predict result as before. The idea is clear and easy to follow. For the theory part, I do not thoroughly check the theoretical proof but the theorem statement sounds reasonable for me. The experiment shows the HAG performs faster in both training and inference.

Generally speaking, I think this paper has good theory analysis, the speed-up effect is also good from the experimental result.  However, I still have some concerns and comments.

1. The algorithm seems hard to apply on the attention-based Graph Neural network, which achieves good performance in several benchmarks these years. In other words, the redundancy of the node aggregate only exists in the Graph Convolution model with the fix node weight, which is replaced by a dynamic weight in many latest models with higher performance. That weakens the empirical use of this algorithm.

2. The authors state that the HAG can optimize various kinds of GNN models, but the experiment only shows the results on a small GCN model. More GNN results in different models and settings would make the algorithm more convincing.

In conclusion, I think this is a good paper. Regards the comments above, I prefer a grade around the borderline.

**Experience Assessment:**

I have read many papers in this area.

**Review Assessment: Checking Correctness Of Derivations And Theory:**

I assessed the sensibility of the derivations and theory.

**Review Assessment: Checking Correctness Of Experiments:**

I assessed the sensibility of the experiments.

**Review Assessment: Thoroughness In Paper Reading:**

I read the paper at least twice and used my best judgement in assessing the paper.

---

> ### Author Response · Authors · 2019-11-15
> **Response to Review#4**
>
> We thank the reviewer for providing a thorough review and asking valuable questions. The reviewer raises a concern about the broad applicability of HAGs. HAG is broadly applicable to the majority of GNN models, including GraphSage, PinSage, GCN, GIN, SGC, DCNN, DGCNN, and many others. HAG provides significant speedups while provably preserving model accuracy (both for training and inference).
>
> ### HAG does not support GAT
> We thank the reviewer for making this point. This is correct and HAG optimization does not apply to graph attention networks (GAT). The HAG graph representation is designed for GNNs with a neighborhood aggregation scheme (formally defined in Algorithm 1), and is applicable to most existing GNN models, including GraphSAGE, PinSAGE, GCN, GIN, SGC, DCNN, DGCNN, and many others. Thus, it is reasonable to conclude that HAG applies to a significant majority of GNN models. Because such GNN-graphs are processed individually in these GNN models, they contain significant redundant computation (up to 84%), and HAG can reduce the overall computation by 6.3x, while provably preserving the original model accuracy.
> We do appreciate the comment, and we have added a paragraph at the end of page 8 to discuss this limitation of HAG in the updated paper.
>
> ### More GNN results in different models would make the paper more convincing
> We thank the reviewer for the constructive feedback. In the updated paper, we have evaluated HAG on more GNN models and observed similar or even better performance improvement. In particular, we have further evaluated HAG on GIN (Xu et al., 2019) and SGC (Wu et al., 2019). The results are shown in Figure 5 on page 12. Compared to the GCN model, our HAG optimizations achieve similar speedups on GIN and better speedups on SGC.

---

### Public Comment · ~Xiaojian_Wu1 · 2019-10-23
**clarification on memory overhead**

In section 5.5, you mentioned "by gradually increasing the capacity,the search algorithm eventually ﬁnds a HAG with 100K aggregation nodes, which consume 6MB of memory (0.1% memory overhead) while improving the training performance by 2.8×."  It seems that a lot of extra nodes are added but only 0.1% memory overhead is introduced, could you explain how this number was calculated?
And how much memory overhead will be introduced for extra edges?

---

> ### Author Response · Authors · 2019-10-23
> **The HAG memory overhead is negligible (~0.1%), and HAG can reduce the number of edges by 1.3-5.6x.**
>
> Thanks for your interests in the paper.
>
> The negligible memory overhead is because the intermediate aggregations nodes do not need to be memorized for back propagation, and our HAG implementation uses the same memory across all layers. Caching the intermediate results for 100K nodes (with 16 activations per node) requires approximately 6MB GPU memory, which is negligible compared to the overall memory usage (~6GB) for training COLLAB.
>
> For the second question, our HAG approach can actually reduce the memory usage for edges, since a HAG contains 1.3x-5.6x fewer edges than the original graph representation. The two bottom charts in Figure 3 show the edge comparison between HAG and the original graph representation.
>
> We will include more analysis on the memory overhead in the revised paper.

---

### Decision · Program_Chairs · 2019-12-19

**Decision:**

Reject

**Comment:**

This paper proposes a new graph Hierarchy representation (HAG) which eliminates the redundancy during the aggregation stage and improves computation efficiency. It achieves good speedup and also provide theoretical analysis. There has been several concerns from the reviewers; authors' response addressed them partially. Despite this, due to the large number of strong papers, we cannot accept the paper at this time. We encourage the authors to further improve the work for a future version.